# Cutting Down Training Memory by Re-fowarding

## Abstract

Deep Neutral Networks(DNNs) require huge GPU memory when training on modern image/video databases. Unfortunately, the GPU memory as a hardware resource is always finite, which limits the image resolution, batch size, and learning rate that could be used for better DNN performance. In this paper, we propose a novel training approach, called **Re-forwarding**, that substantially reduces memory usage in training. Our approach automatically finds a subset of vertices in a DNN computation graph, and stores tensors only at these vertices during the first forward. During backward, extra local forwards (called the Re-forwarding process) are conducted to compute the missing tensors between the subset of vertices. The total memory cost becomes the sum of (1) the memory cost at the subset of vertices and (2) the maximum memory cost among local re-forwards. Re-forwarding trades training time overheads for memory and does not compromise any performance in testing. We propose theories and algorithms that achieve the optimal memory solutions for DNNs with either linear or arbitrary computation graphs. Experiments show that Re-forwarding cuts down up-to $80\%$ of training memory on popular DNNs such as Alexnet, VGG, ResNet, Densenet and Inception net.

## 1 Introduction

The standard DNN training process consists of two alternated stages: forward and backward. Fig. 1 (a) illustrates an example of feed-forward neural networks. In the forward stage, the network takes an input tensor, $[BatchSize \times Channel \times Width \times Height]$, and computes the tensors at each layer until producing the output. In the backward stage, difference between the output and ground truth is passed back along the network to compute the gradients at each layer. The regular training approach saves tensors at all layers during forward, because they are all needed to compute gradients during backward. The total memory cost is the sum of cost over all layers.

In popular backbone DNNs for feature extraction of images, such as AlexNet (Krizhevsky et al. (2012)), VGG (Simonyan & Zisserman (2014)) and ResNet (He et al. (2016)), the memory cost increases quadratically with the input image resolution and network depth. For example, given an median size input tensor of $(32, 3, 224, 224)$, ResNet101 requires around 5000 MB. In more challenging tasks, DNNs that detect small objects and large number of object categories require input image resolution of more than $600 \times 600$ (Ren et al. (2015); Singh et al. (2017); Redmon & Farhadi (2018)). The memory issue is worse for video-based DNNs, such as CDC (Shou et al. (2017)), C3D (Ji et al. (2013)) and 3D-ResNet (Hara et al. (2017)). To model complex activities in video, the input tensor may contain 64 frames. Moreover, DNN training takes much more memory than testing. In order to train DNNs with large databases and big learning rate, the batch size can be up to 64. In training DNN compositions, such as Generative adversarial networks (GANs), multiple generator and discriminator networks are simultaneously stored in GPU memory.

Existing efforts to address memory issues presented three main approaches: (1) Better single GPUs. Recent GPUs provide larger memory at the expense of exponentially growing price and power consumption. For instance, from TitanXp, Quadro P6000 to Tesla V100, for 1-2.7 times increase in memory, the prices increase 2.8-8.5 times. (2) Parallelization among multiple GPUs (Dean et al. (2012); Shi et al. (2009); Langford et al. (2009); Mcdonald et al. (2009); McDonald et al. (2010); Zinkevich et al. (2010); Agarwal et al. (2014); Agarwal & Duchi (2011)), which requires expensive

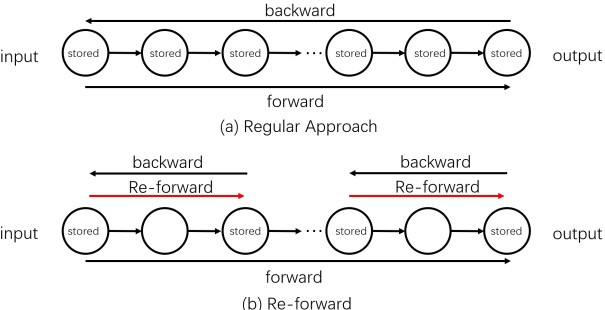

Figure 1: Regular Training Approach vs. Re-forwarding (our). (a) The regular approach saves all tensors during forward, and uses these tensors to compute gradients during backward. (b) Re-forwarding (our) saves a subset of tensors during the first forward, and conducts "Re-forward" to compute tensors for gradients during backward.

clusters, introduces substantial I/O cost, and does not reduce the total memory cost. (3) Low-level heuristic techniques. Optimization of computation graphs (Aho et al. (1986)), which merges inplace operations into non-inplace operations to cut down memory. Liveness analysis (Aho et al. (1986)), which dynamically recycles garbage tensors in training epochs. These approaches are specific to certain DNN structures, data and tasks.

To address above issues, we propose a fundamental approach that explores trade-off between memory and computation power of GPUs. Note that recent affordable GPUs, although limited in memory ( 12GB), provide exceptional improvement in GPU cores and FLOPS. Trading computational time for memory is a very attractive solution that make it possible to train very heavy DNNs with finite GPU memory. Our approach only saves tensors at a subset of layers during the first forward, and conduct only extra *local forwards* to compute the missing tensors needed during backward. We call the extra forward process as **Re-forwarding**. The total memory cost is the sum of (1) the cost at the subset of layers and (2) the maximum memory cost among local re-forwards. Training with Re-forwarding, see Fig. 1 (b), leads to substantial memory reduction. We propose sophisticate theories and efficient algorithms that achieve the optimal memory solution of *arbitrary* computation graphs.

## 2 RELATED WORK

To alleviate the memory pressure from a single GPU processor, many researchers utilized the well-established techniques for distributed computation (Dean et al. (2012); Shi et al. (2009); Langford et al. (2009); Mcdonald et al. (2009); McDonald et al. (2010); Zinkevich et al. (2010); Agarwal et al. (2014); Agarwal & Duchi (2011)). These techniques distribute memory pressure to possibly infinite GPUs or server clusters, but do not reduce the total memory cost of DNNs.

Other researchers reduced the memory on finite hardware by optimizing computation graph of DNN and performing liveness analysis. The computation graph of DNNs describes the dependencies of tensors among layers. Liveness analysis recycles garbage to manage memory. These ideas were originated from compiler optimization (Aho et al. (1986)) and has been widely adopted by deep learning frameworks: Theano (Bastien et al. (2012); Bergstra et al. (2010)), MXNet (Chen et al. (2015)), Tensorflow (Abadi et al. (2016)) and CNTK (Yu et al. (2014)). Some other techniques efficiently swap data between CPU and GPU (Wang et al. (2018); Rhu et al. (2016)). These techniques usually cost extra I/O time and still do not actually reduce the total memory cost.

The closest work to our approach, Chen et al.(Chen et al. (2016)), uses the gradient checkpoints (similar to the subset of layers in Re-forwarding). However, **(Chen et al. (2016)) only worked on linear computation graph** via a heuristic algorithm. Our approach generates optimal solutions for **both linear and arbitrary computation graphs**. Our algorithm reduces training memory by manipulating high-level tensors, therefore is generalizable to any DNNs and their compositions. All previous techniques are compatible to our approach and can further improve the memory efficiency of DNN training.

## 3 LINEAR COMPUTATION GRAPH (LCG)

Denote a computation graph as $G = (E, V)$. $E = \{e_i\}$ and $V = \{v_i\}$ are the edges and vertices in the computation graph, respectively. In deep neural networks, the vertices represent the tensors and the edges represent operations. Denote function $l(\cdot)$ as a measure of memory cost. $V_R$ is the subset of vertices saved during the first forward. $l(v_i)$ is defined as the memory cost of storing vertex $v_i$. For two adjacent vertices $v_i$ and $v_j$ in set $V_R$, the memory cost during re-forwarding from $v_i$ to $v_j$ is defined as $l(v_i, v_j) = \sum_{t=i+1}^{j-1} l(v_t)$, which is the sum of cost over all the vertices between $v_i$ and $v_j$. Using these notations, the memory cost of training with re-forwarding is formulated as

$$\min_{V_R} \sum_i l(v_i) + \max_j l(v_j, v_{j+1}), \tag{1}$$

where the first term is the sum of the memory cost of all the stored tensors, and the second term is the maximal cost among the re-forwards.

For easy illustration, we start by formulating Re-forwarding on Linear Computation Graphs (LCG) (Fig. 2 (a)). For LCGs, Eqn. 1 can be solved in two cases.

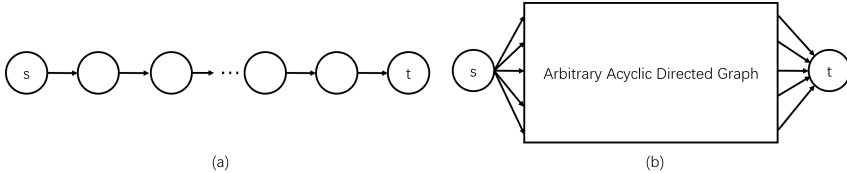

(a)                  (b)

Figure 2: (a) Linear Computation Graph (LCG). "s" denotes the start vertex,"t" denotes the end vertex. (b) Arbitrary Computation Graph (ACG). The structure between "s" and "t" vertices may contain arbitrary branches and connections.

**Case(1) LCG with Identical Vertex Cost:** Suppose a LCG has $N$ vertices, each of which has the same cost $l(v_i) = \frac{1}{N}$ and the total cost is 1. Obviously, the optimal solution is reached when vertices in $V_R$ are distributed evenly in the LCG. Suppose the number of vertices in $V_R$ is $k$. The total cost is then $\frac{k}{N} + \frac{1}{k}$. The optimal solution of Eqn. 1 is $k = \sqrt{N}$, and the optimal total cost is $\frac{2}{\sqrt{N}}$.

**Case (2) LCG with Non-identical Vertex Cost:** When the assumption of identical cost does not hold, the solution to Eqn. 1 does not have an analytic form. Denote the maximal Re-forward cost $\max_j l(v_j, v_{j+1})$ as a constant $C$, and the solution to Eqn. 1 is reduced to solving for $\min_{V_R} \sum_i l(v_i)$.

---

**Algorithm 1** Linear Computation Graph (LCG) Solver

---

1: **for** each vertex pair $(v_i, v_j)$ in $G$ **do**
2:  Set the maximal term as $l(v_i, v_j)$
3:  Construct Accessibility Graph
4:  Find the shortest path in the Accessibility Graph as the solution
5:  Compute the actual total cost of the solution
6:  Save the solution if it's better.
7:  Suppose the actual max term of this solution is $B$, and $l(v_i, v_j) = C$, skip the loops where $B \le l(v_i, v_j) < C$

---

All the Re-forward costs in an optimal solution satisfy the constraint $l(v_j, v_{j+1}) \le C$. We solve Eqn. 1 by constructing a new graph, called Accessibility Graph $G^A = (E^A, V)$. The edges of $G^A$, called Accessibility Edge $e_{ij}^A$, exists between vertex $v_i$ and $v_j$ if and only if $l(v_i, v_j) \le C$. Now the problem of solving $\min_{V_R} \sum_i l(v_i)$ is equivalent to finding the shortest path from the source vertex and the target vertex in the Accessibility Graph. Notice that in the optimal solution, the max term equal the one maximal term among all $l(v_i, v_{i+1})$ terms. To traverse all possible max terms, we can simply compute the loss of every vertex pair and use it as a possible max term. Given a max term $C$, suppose the actual max term of the solution under $C$ is $B$ and $B < C$. It's obvious that for all the max terms $B \le max < C$, the solution would be the same solution. Therefore, these max terms can be skipped. **Algorithm 1** summarizes the process for searching an optimal solution for

LCG. Suppose there are $N$ vertices in the computation graph, the time complexity of **Algorithm 1** is $O(N^4)$[1].

## 4 ARBITRARY COMPUTATION GRAPH(ACG)

As generalization of DNNs with LCG, we present theory[2] and algorithms for DNNs with Arbitrary Computation Graphs (ACG), in particular the acyclic directed graphs(Fig. 2 (b)).

### 4.1 ASSUMPTION FOR OPTIMALITY

The optimal solution of Re-forwarding corresponds to an optimal division of ACG, such that memory cost (Eqn. 1) is minimum. We denote that an ACG is divided into end-to-end segments by a set of vertices. These end-to-end segments can have multiple endpoint vertices, for example, multiple source vertices and multiple target vertices. In this paper, as an assumption and also for simplification, these end-to-end segments are narrowed down to those with only one source vertex and one target vertex.

Another assumption in the case of ACG is imposed on the operation that has multiple inputs: one can compute the gradients of output with respect to the gradients of inputs without using the current value of inputs. Examples of operations that meet this assumption are: concatenation (the gradient of output is also the concatenation of the gradient of input), add (the gradient of output equals the gradient of input), etc. An example that breaks this assumption is multiplication (the gradient of input depends on the input). Fortunately, most of the popular networks meet this assumption. A simple way to remove this assumption is to store all the input tensors of this multi-input operation. However, this is not modeled by our loss function and may lead to sub-optimal solution.

In summary, there are only two assumptions in our approach: (1) the segment in a solution only has two endpoints (source and target). (2) the multi-input operation can compute the gradients of output without using the current value of input. Under these two assumptions, our approach is optimal for ACGs.

### 4.2 DEFINITION AND THEOREM

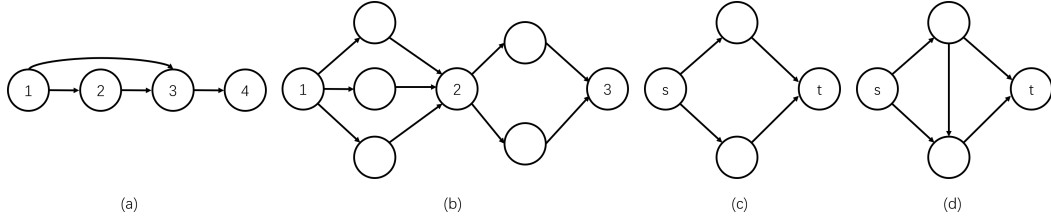

(a)  (b)  (c)  (d)

Figure 3: Closed Set Examples: (a) Closed set in a graph. there cannot exist a closed set between $v_2$ and $v_4$ because $v_3$ depends on $v_1$. There can exist a closed set between $v_1$ and $v_3$ because $v_2$ doesn't depend on any other vertex. (b) Splittable Closed Set (Type 1). $v_2$ is the splitting vertex of $s_{13}$. (c) Branched Closed Set (Type 2). (d) Non-branched Closed Set (Type 3).

**Definition 1.** *Closed Set: A set $s$ containing vertices and edges is a closed set if and only if it satisfies the following three properties: 1. All the vertices of $s$ have a common ancestor $v_i$ and a common descendent $v_j$; 2. Denote the vertex subset of $s$ as $V$, edge subset as $E$, and the set of edges between two arbitrary vertices of $V \cup \{v_i, v_j\}$ is $E'$, the edge from $v_i$ to $v_j$ (if exists) as $e_{ij}$. $E$ must either be $E'$ or $E' - \{e_{ij}\}$; 3. An arbitrary $v_1 \in V$ doesn't have edge with another arbitrary $v_2 \notin V \cup \{v_i, v_j\}$. For multiple valid closed sets between $v_i$ and $v_j$, we denote the largest one as $s_{ij}$*

**Definition 2.** $[s_{ij}] = s_{ij} \cup \{v_i, v_j\}$. $[s_{ij}) = s_{ij} \cup \{v_i\}$. $(s_{ij}] = s_{ij} \cup \{v_j\}$

In the definition of **Closed Set**, property 1 corresponds to the two endpoint assumption in section 4.1 where the two endpoints become $v_i$ and $v_j$ in the definition. Property 2 confines the edge subsets of

---

[1]More detailed complexity analysis is in the appendix due to space limitation.
[2]All proofs are in the appendix due to space limitation.

$s$ to be one of two cases: $E'$ or $E' - \{e_{ij}\}$. Both cases are valid although they have different edges. Property 3 guarantees the independence of such a set $s$, meaning that the vertices within $s$ have no connections with other vertices outside $s \cup \{v_i, v_j\}$. As there might be multiple valid closed sets between $v_i$ and $v_j$, which corresponds to the **Branched Closed Set** in Definition 5, we denote the largest closed set between $v_i$ and $v_j$ as $s_{ij}$ and denote smaller closed set with an extra superscript, such as $s_{ij}^1$.

**Definition 3. *Splitting Vertex:*** *A vertex $v_t \in s_{ij}$ is a splitting vertex of $s_{ij}$ if and only if $s_{it}$ exists, $s_{tj}$ exists and $s_{ij} = s_{it} \cup s_{tj} \cup \{v_t\}$ and $s_{it} \cap s_{tj} = \emptyset$*

**Definition 4. *Splittable Closed Set (Type 1):*** *closed set with at least 1 splitting vertex.*

The definition of **Splitting Vertex** is to describe whether a closed set can be divided into two linearly arranged closed set. A closed set is splittable if it has at least 1 splitting vertex and is defined as **Closed Set Type 1**.

**Definition 5. *Branched Closed Set (Type 2):*** *A closed set is branched if it has 0 splitting vertex and can be divided into branches: $s_{ij} = s_{ij}^1 \cup s_{ij}^2$ and $s_{ij}^1 \cap s_{ij}^2 = \emptyset$*

**Definition 6. *Non-branched Closed Set (Type 3):*** *A closed set $s_{ij}$ is non-branched if it has 0 splitting vertex and no branch: $\nexists s_{ij}^1 \subsetneq s_{ij}$*

Among closed set with no splitting vertex, we categorize closed set with branches as **Closed Set Type 2**, and closed set without branches as **Closed Set Type 3**. The examples of different types of closed set are shown in Fig. 3.

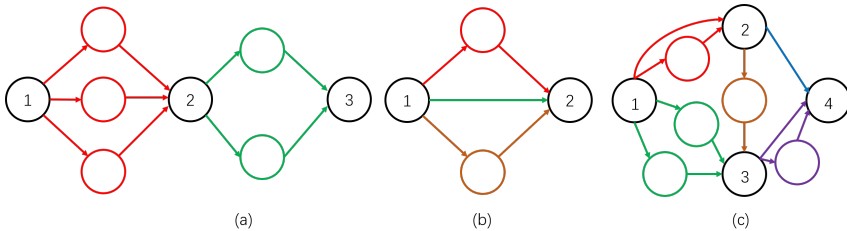

(a)   (b)   (c)

Figure 4: Example divisions of three types of closed sets. Members of a division are colored differently. (a) Division of closed set type 1. The division is $\{[s_{12}], [s_{23}]\}$ (b) Division of closed set type 2. The division is $\{[s_{12}^1], [s_{12}^2], [s_{12}^3]\}$ (c) Division of closed set type 3. The division is $\{[s_{12}], [s_{13}], [s_{23}], [s_{24}], [s_{34}]\}$

**Definition 7. *Maximal Split:*** $\{[s_{pq}]\}$ *is a maximal split of non-branched $s_{ij}$ if $[s_{ij}] = \cup\{[s_{pq}]\}$ and $\forall s_{ab}, s_{cd} \in \{[s_{pq}]\}, s_{ab} \cap s_{cd} = \emptyset$ and $\nexists\{[s'_{pq}]\} \subsetneq \{[s_{pq}]\}$ such that $\cup\{[s'_{pq}]\} = [s_{kt}] \subsetneq [s_{ij}]$*

**Definition 8. *Division of Closed Set:*** *For type 1, its division is the linear segments separated by all its splitting vertices; for type 2, its division is all its branches, any of which cannot be divided into more branches; for type 3, its division is its maximal split.*

For closed set type 1, it can be divided into linearly arranged segments. For closed set type 2, it can be divided into branches. So here we investigate the division of closed set type 3. As we don't want trivial division, for example, division that is formed by every edge in the closed set, we define **Maximal Split** to describe the split such that each member of the split is as large as possible. An example of maximal split is shown in Fig. 4 (c). In the definition of maximal split, the term maximal is implied by saying that any subset of this split cannot be combined into a single closed set. If it can, then the maximal split will be formed by this larger closed set and all the rest of the previous split. For closed set type 3, we use its maximal split as its division.

**Definition 9. *Division Tree:*** *Division tree is a representation of a computation graph, where the root node is the whole computation graph, the leaf nodes are all the single tensors in the computation graph, and for a non-leaf node, its children is the members of its division.*

With the division of 3 types of closed sets, the computation graph can be reorganized into a division tree (Figure 5) where a non-leaf node would be a closed set and its children would be its corresponding division. The root node is the whole computation graph, the largest closed set, and the leaf nodes would be single tensors in the computation graph. With division tree, we can apply divide-and-conquer to search for optimal solution.

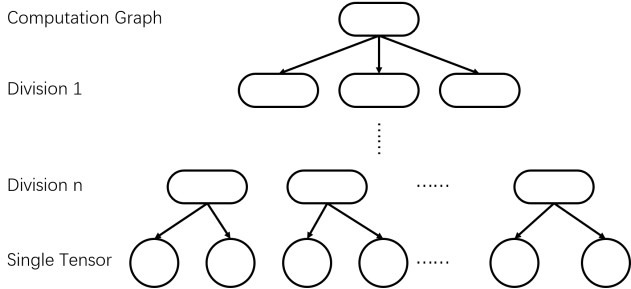

Figure 5: In this tree, the root node is the whole computation graph. All the leaf nodes are single tensors. Every other node except root and leaves is a member of the division of its parent.

**Theorem 1.** *The division tree of a computation graph is unique and complete.*

The uniqueness of the division tree indicates that the optimal solution of the division tree would also be the optimal solution of the whole computation graph. The completeness indicates that the division tree has included all the possible members of solution and represents the whole search space for the optimal solution. Theorem 1 is proved in the appendix.

### 4.3 ALGORITHM

We search optimal solutions for ACGs by solving several sub-problems using Algorithm 2-4 respectively. Based on these components, we present our final solver as Algorithm 5.

**Algorithm 2** judges whether a vertex is a splitting vertex of a closed set. This algorithm mainly follows the Definition 3 and uses vertex set to check the property of a splitting vertex. With this algorithm, we can judge whether a closed set is type 1 and get its division if it is. Suppose there are $N$ vertices in $s_{ij}$, the time complexity of **Algorithm 2** is $O(N^2)$.

---

**Algorithm 2** Judge whether a vertex $v_t$ is a splitting vertex of closed set $s_{ij}$

---

1: Let $\{v_{in}\}$ be the vertices of all the vertices within $[s_{ij}]$ that have paths to $v_t$. Let $\{v_{out}\}$ be the vertices of all the vertices within $[s_{ij}]$ that have paths from $v_t$.
2: **if** $\{v_{in}\} \cup \{V_{out}\} \cup \{v_t\} = \{v | v \in [s_{ij}]\}$ and $\{v_{in}\} \cap \{V_{out}\} = \emptyset$ and $\nexists v_1 \in \{v_{in}\}, v_2 \in \{v_{out}\}$, $v_1, v_2$ have connections **then**
3:     Return true
4: **else**
5:     Return False

---

**Algorithm 3** examines whether a closed set is branched. It uses a growing algorithm to check whether an independent subpart of this closed set can form a closed set. If a non-trivial closed set $s_{ij}$ has an edge from $v_i$ to $v_j$, then it's branched because this edge itself can be treated as a closed set. Combined with Algorithm 2, we can know the type of a closed set and get its division if it's type 2. Suppose there are $N$ vertices in $s_{ij}$, the time complexity of **Algorithm 3** is $O(N^2)$.

**Algorithm 4** addresses the problem of finding the maximal split, the division of a closed set type 3 $s_{ij}$. First get all the possible closed sets within $s_{ij}$ and use a property of maximal split to judge whether this closed set is a member of the maximal split. The property is: there cannot exist another closed set $s_{ab} \subsetneq s_{ij}$ but contains any member of this maximal split. This property is proved in Lemma 6 of the appendix. Suppose there are $N$ vertices in $s_{ij}$, the time complexity of **Algorithm 4** is $O(N^4)$.

**Algorithm 5** is the solver for ACGs. First, the division tree of the computation graph is built. Similar to the linear solver, a max term list is formed by the cost of all the possible closed sets for traverse. Given a max term, we propose a greedy idea: for a closed set, never expand it unless the its cost exceed the max term. In other word, if the max term doesn't allow a leap over this closed set, we expand it, otherwise, do not expand it. Because once expanded, some cost of other vertices inside this closed set might be introduced, and the cost will never be smaller than unexpanded. If some children of the closed set type 1 are expanded, the rest reforms a few linear segments and still can be solved by the linear solver. If some children of the closed set type 2 or 3 are expanded, the other

---

**Algorithm 3** Judge whether $s_{ij}$ is branched

---

1: **if** $s_{ij}$ has at least 1 vertex **then**
2:    **if** $s_{ij}$ includes an edge from $v_i$ to $v_j$ **then**
3:       Return true
4:    **else**
5:       Initialize a vertex set $s = \{v_k\}$. $v_k \in s_{ij}$ is a randomly chosen vertex.
6:       **while** True **do**
7:          For any $v_t \in s_{ij}, v_t \notin s$ that has connection to any $v_k \in s$, add $v_t$ to $s$.
8:          **if** No more vertex can be added to $s$ **then**
9:             Break
10:       **if** $s = \{v \in s_{ij}\}$ **then**
11:          Return false
12:       **else**
13:          Return true
14: **else**
15:    Return false

---

**Algorithm 4** Find the maximal split of a non-branched $s_{ij}$ with 0 splitting vertex

---

1: **for** each vertex pair $(v_k, v_t)$ except $(v_i, v_j)$ in $[s_{ij}]$ **do**
2:    For all the vertices $\{v\}$ that have paths from $v_k$ and have paths to $v_t$.
3:    **if** $\nexists v_2 \notin \{v\}$ and $v_2 \neq v_k, v_t$, $v_2$ has connection to a $v_1 \in \{v\}$ **then**
4:       Form a closed set $s_{kt}$ with all these vertices.
5: **for** each formed closed set $s_{kt}$ **do**
6:    If there doesn't exist a $s_{ab}$ such that $s_{kt} \subsetneq s_{ab} \subsetneq s_{ij}$, put $s_{kt}$ into the maximal split.

---

members remain unexpanded and need no changes. Suppose there are $N$ vertices in computation graph, the time complexity of **Algorithm 5** is $O(N^4)$.

---

**Algorithm 5** Arbitrary Computation Graph (ACG) Solver

---

1: Get all possible closed set and their costs. Use their costs to form the max term list.
2: Reorganize the computation graph into a division tree: from the root node (the computation graph), build its children from its division, until all the leaf nodes are single tensors.
3: **for** each possible max term $m$ in max term list $\{m\}$ **do**
4:    **if** current closed set is type 1 **then**
5:       For all the children that have cost larger than current max term. Expand them and solve the next level.
6:       All the expanded children have separated the current closed set to linear segments. Solve all the linear segments with current max term.
7:    **else**
8:       For all the children that have cost larger than current max term. Expand them and solve the next level.
9:       All the other members remain unexpanded.
10:    Summarize the total loss, save the current solution if it's better.

---

## 5 EXPERIMENT

We evaluated Re-forwarding on two main groups of neural networks (1) networks with linear structures, such as Alexnet (Krizhevsky et al. (2012)) and vgg series (Simonyan & Zisserman (2014)). (2) networks with non-linear structures, such as Resnet series (He et al. (2016)), Densenet series (Huang et al. (2017)) and Inception net (Szegedy et al. (2016)). For each network in Table 1, an computation graph is built such that every vertex is a Float32 tensor, every edge is an operation, and the memory cost of a vertex is its tensor size (measured in MB). We compared Re-forwarding with Chen (Chen et al. (2016)) and the regular training approach. **Note that Chen et al. (2016) only worked on linear computation graphs**. To compare with (Chen et al. (2016)) on non-linear networks, we **manually** re-organized all the non-linear computation graphs into linear computation graphs with their splitting vertices, and fed them to Chen et al. (2016) (see Table 1 "Chen et al. (2016) manual

Table 1: Training memory usage and time overhead of the regular, Chen et al. (2016), Chen et al. (2016) manual and Re-forwarding (ours) approach on linear and non-linear computation graph.

| Linear network | Regular (MB) | Chen et al. (2016) (MB) | Re-forwarding (ours) (MB) | Memory Cut off (ours) | Regular Training Time (s) | Space Efficient Training Time (s) | Time Overhead |
|---|---|---|---|---|---|---|---|
| Alexnet batch 1024 | 3550 | 3108 | **2620** | 26% | 1.295 | 1.816 | 40% |
| Vgg11 batch 64 | 2976 | 2292 | **1802** | 39% | 0.606 | 0.819 | 35% |
| Vgg13 batch 64 | 4152 | **2586** | 2586 | 38% | 1.020 | 1.333 | 31% |
| Vgg16 batch 64 | 4470 | 2894 | **2586** | 42% | 1.307 | 1.696 | 30% |
| Vgg19 batch 64 | 4788 | 2894 | **2502** | 48% | 1.593 | 2.060 | 29% |
| Non-linear network | Regular (MB) | Chen et al. (2016) manual (MB) | Re-forwarding (ours) (MB) | Memory Cut off (ours) | Regular Training Time (s) | Space Efficient Training Time (s) | Time Overhead |
| Resnet18 batch 256 | 5402 | **2898** | **2898** | 46% | 1.144 | 1.599 | 40% |
| Resnet34 batch 128 | 3900 | 1936 | **1544** | 60% | 1.041 | 1.419 | 36% |
| Resnet50 batch 64 | 5206 | 2332 | **1798** | 65% | 0.740 | 1.027 | 40% |
| Resnet101 batch 32 | 3812 | 1216 | **970** | 75% | 0.624 | 0.853 | 37% |
| Resnet152 batch 16 | 2810 | 636 | **564** | 80% | 0.450 | 0.628 | 39% |
| Densenet121 batch 32 | 3984 | 1012 | **776** | 81% | 0.558 | 0.789 | 42% |
| Densenet161 batch 16 | 3658 | 744 | **616** | 83% | 0.511 | 0.708 | 39% |
| Densenet169 batch 32 | 4826 | 998 | **848** | 82% | 0.714 | 1.022 | 43% |
| Densenet201 batch 16 | 3164 | 600 | **582** | 82% | 0.449 | 0.651 | 45% |
| Inceptionv3 batch 32 | 2976 | 1026 | **910** | 69% | 0.563 | 0.763 | 35% |
| CustomNet batch 64 | 3233 | Not Applicable | **1353** | 58% | 1.226 | 1.648 | 34% |

(MB)"). Our Re-forwarding approach directly works on arbitrary computation graphs. We have also included a customized network ("CustomNet"), on which even the manual version of Chen's approach is not applicable. **Our approach directly works on all networks.** The computation graph of this network is visualized in the appendix.

All experiments were conducted in Pytorch. To remove irrelevant GPU memory cost, such as model and Pytorch CUDA interface cost, all training memory costs were measured with two different input sizes and compute the difference between two measurements. For example, to measure the memory cost of Alexnet with input size $[BatchSize, Channel, Width, Height] = [16, 3, 224, 224]$, we first record the training memory of input $[16, 3, 224, 224]$ as $r_1$, and input $[32, 3, 224, 224]$ as $r_2$. The actual memory cost given $[16, 3, 224, 224]$ input is measured as $r_2 - r_1$. To use existing DNN implementations, the input of Inception net is $[BatchSize, 3, 300, 300]$, and the input of all other networks is $[BatchSize, 3, 224, 224]$. We also measure the training time (time of 1 training iteration) for the regular approach and our approach. Each time is measured as the average of 20 iterations. Our approach has the same training time as Chen's approach and its manual version, see "Space Efficient Training Time" in Table 1.

Table. 1 shows that Re-forwarding cuts down huge amount of memory from the regular approach at reasonable time overheads: 26% space off and 40% time overhead for Alexnet, around 40% space off and 40% time overhead for Vgg series. For Resnet series, the deeper network, the more memory was cut down. On the deepest Resnet152, 80% space off was achieved with only 39% time overhead. For Densenet series, more than 80% space off was achieved with around 40% time overhead. Notice that, Chen et al. (2016) only works on linear networks. Its results on non-linear networks were **manually synthesized**. Re-forwarding directly works on non-linear networks and constantly outperformed Chen et al. (2016) and its "manual" version. This supports our claim that Re-forwarding is optimal.

# 6 CONCLUSION

Re-forwarding is a fundamental approach that explores trade-off between memory and computation power of GPUs. By saving tensors at a subset of layers during forward, and conducting extra *local forwards* for backward, Re-forwarding makes it possible to train very heavy DNNs with finite GPU memory. To our knowledge, our theoretical and algorithmic results are the first top-down work that achieve an optimal memory solution for arbitrary computation graphs in DNNs. Re-forwarding can be further embedded and optimized with any low-level techniques such as distributed computing, GPU/CPU swapping, computation graph optimization and liveness analysis.

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

## A    PROOF

### A.1    LEMMAS

**Lemma 1.** *If $s_{ij} \cap s_{kt} \neq \emptyset$ and $s_{ij} \not\subset s_{kt}$ and $s_{kt} \not\subset s_{ij}$, then $s_{ij} \cap s_{kt} = s_{kj}$ or $s_{ij} \cap s_{kt} = s_{it}$*

*Proof.* Let $[s_{ij}] \cap [s_{kt}] = s = \{v, e\}$. Let $v_p$ be the source vertex of $s$, $v_q$ be the target vertex of $s$. If $v_p \neq v_i$ and $v_p \neq v_k$ and $v_q \neq v_j$ and $v_q \neq v_t$, then $v_i, v_k$ has path to $v_p$ and $v_j, v_t$ has path from $v_q$. Therefore, $v_p$ has at least 2 immediate parents $v_a, v_b$ with $v_a \in [s_{ij}], v_a \notin [s_{kt}], v_b \in [s_{kt}], v_b \notin [s_{ij}]$. If so, the independence of $s_{ij}$ and $s_{kt}$ is violated. Therefore, $v_p$ must be $v_i$ or $v_k$.

Same on $v_q$, $v_q$ must be $v_j$ or $v_t$.

If $v_p = v_i, v_q = v_j$, then $s_{ij} \subset s_{kt}$. If $v_p = v_k, v_q = v_t$, then $s_{kt} \subset s_{ij}$. Therefore, $v_p = v_i, v_q = v_t$ or $v_p = v_k, v_q = v_j$. Suppose $v_p = v_k, v_q = v_j$, let's prove $s$ is a closed set.

With $s \subset s_{kt}, \forall v_1 \in s$, $v_1$ has no edge with $v_2 \notin [s_{kt}]$. With $s \subset s_{ij}, \forall v_1 \in s$, $v_1$ has no edge with $v_2 \notin [s_{ij}]$. Therefore, $\forall v_1 \in s$, $v_1$ has no edge with $v_2 \notin [s]$. The independence of $s$ is guaranteed.

In the discussion before, we can see the source vertex $v_p$ of $s$ must be either $v_i$ or $v_k$. If $v_i$ and $v_k$ are both the source vertices of $s$, then $v_i \in [s_{kt}]$ and $v_k \in [s_{ij}]$, $v_i$ has path to $v_k$ and $v_k$ has path to $v_i$, which will force $v_i = v_k$ because the $s_{ij}, s_{kt}$ is acyclic. Same on $v_q$, $s$ can only have 1 source vertex and 1 target vertex. Therefore, $s$ is closed set.

Therefore, $s_{ij} \cap s_{kt} = s_{kj}$ or $s_{ij} \cap s_{kt} = s_{it}$. □

**Lemma 2.** *The intersection of two closets $s = s_i \cap s_j \neq \emptyset$ is also a closet*

*Proof.* Given the independence of $s_i$ and $s_j$, the independence of $s$ is obvious. The remaining thing is whether $s$ only has 1 source vertex and 1 target vertex. In the proof of Lemma 1, we can see any source or target vertex of $s$ will eventually become source or target vertex of $s_i$ and $s_j$. With simple discussion, we can have this lemma. □

**Lemma 3.** *If $s_{ij} \cap s_{kt} = s_{kj} \neq \emptyset$, then $v_k$ is the splitting vertex of $s_{ij}$ and $v_j$ is the splitting vertex of $s_{kt}$*

*Proof.* Let's first prove that $v_k$ is the splitting vertex of $s_{ij}$.

Let $s = s_{ij} - [s_{kj})$. Obviously, $s_{ij} = s \cup s_{kj} \cup \{v_k\}$ and $s \cap s_{kj} = \emptyset$. We only need to prove that $s$ is closed set. For convenience, let's denote $[s] = s \cup \{v_i, v_k\}$

$v_i$ is obviously the only source vertex of $[s]$ because $v_i$ is source vertex of $[s_{ij}]$. We discuss the target vertex here. If $v_k$ is not the target vertex of $[s]$, as $v_k \in [s]$, $v_k$ must have path to the target vertex $v$ of $[s]$ and $v$ also has path to $v_j$ as $v \in s_{ij}$. Because $v \notin [s_{kj}]$, in the path from $v$ to $v_j$, there exists an edge that connects a vertex $v_1 \in s$ with a vertex $v_2 \in s_{kt}$ which violates the independence of $s_{kt}$. Therefore, the target vertex of $[s]$ can only be $v_k$.

As $s \subset [s_{ij}), \forall v_1 \in s$, $v_1$ has no edge with $v_2 \notin [s_{ij})$. As $s_{kj}$ is close, $\forall v_1 \in s$, $v_1$ has no edge with $v_2 \in s_{kj}$. $\forall v_1 \in s$, $v_1$ can only have edge with $v_2 \in [s]$. Thus the independence of $s$ is guaranteed. Therefore, $s$ is closed set, $v_k$ is the splitting vertex of $s_{ij}$.

Same on $v_j$, $v_j$ is the splitting vertex of $s_{kt}$ □

**Lemma 4.** *If $s_{ij}$ has $n$ splitting vertices $\{v_1, v_2, ..., v_n\}$, then $s_{ij} = s_{i1} \cup s_{12} \cup ... \cup s_{nj} \cup \{v_1, v_2, ..., v_n\}$*

*Proof.* If $n = 2$, the splitting vertices are $v_1, v_2$, $s_{ij} = s_{i1} \cup s_{1j} \cup \{v_1\} = s_{i2} \cup s_{2j} \cup \{v_2\}$. Let $v_1 \in s_{i2}, v_1 \neq v_2$, then $s_{1j} \cap s_{i2} = s_{12} \neq \emptyset$. According to Lemma 3, $v_1$ is splitting vertex of $s_{i2}$ and $v_2$ is splitting vertex of $s_{1j}$. Therefore, $s_{ij} = s_{i1} \cup s_{12} \cup s_{2j} \cup \{v_1, v_2\}$.

For $n > 2$, the lemma can be proved by repetitively using the conclusion in $n = 2$.

□

**Lemma 5.** *If the non-branched $s_{ij}$ has a maximal split $\{[s_{pq}]\}$, and $|\{[s_{pq}]\}| > 2$, denote $\{v\}$ as all the endpoint vertices of $[s] \in \{[s_{pq}]\}$. Then $\forall v \in \{v\}, v \neq v_i, v_j$, $v$ is the endpoint vertex of at least 3 members of the maximal split.*

*Proof.* If $v_b$ is the endpoint vertex of only 2 members of the maximal split, suppose the 2 members are $s_{ab}$ and $s_{bc}$. If so, $s_{ab}$ and $s_{bc}$ can be merged into $s_{ac}$. If $s_{ac} \neq s_{ij}$, this violates the definition of maximal split. Otherwise, it violates the condition that $s_{ij}$ is non-branched and $|\{[s_{pq}]\}| > 2$. It is impossible that the 2 members are $s_{ab}$ and $s_{cb}$ because in this way $v_b$ has no path to $v_j$ and violates the definition of closed set. If $v_b$ is the endpoint vertex of only 1 member of the maximal split, then $v_b$ must be either $v_i$ or $v_j$. Therefore, this lemma is proved.

$\square$

**Lemma 6.** *Any member of a maximal split can not be the subset of another closed set $s \subsetneq s_{ij}$.*

*Proof.* Suppose the source vertex of $s$ is $v_1$ and target vertex is $v_2$, a member $s_{xy}$ of the maximal split is inside $s$.

Suppose a member $s_{ab}$ of the maximal split has its source vertex $v_a$ inside $s$ and target vertex $v_b$ outside $s$. Then the boundary vertex (the vertex that has edges to the non-overlapping parts of both sets) must be $v_2$, otherwise the independence of $s$ will be violated. Notice that $v_2$ is inside $s_{ab}$ and the independence of $s_{ab}$ needs to be guaranteed, for $\forall v_p \in s, v_p \notin s \cap s_{ab}, v_q \in s \cap s_{ab}$, $v_p$ has no edge with $v_q$. Therefore, $v_a$ is a splitting vertex of $s$.

Similarly, if $s_{ba}$ has its target vertex $v_a$ inside $s$ and source vertex $v_b$ outside $s$, the boundary vertex must be $v_1$ and $v_a$ is a splitting vertex of $s$.

For the closed set $s$, from the discussion above, we know that there are at most 2 members of the maximal split that can overlap with $s$. Other members must be either completely inside $s$ or completely outside $s$. Let's discuss the number of members that overlaps with $s$.

If there are 0 member that overlaps with $s$, $s$ is the union of a subset of members of the maximal split, which violates the definition of maximal split.

If there is 1 member that overlaps with $s$, suppose the corresponding splitting vertex is $v_b$, and the boundary vertex is actually $v_2$. Then $s_{1b}$ is a closed set containing $s_{xy}$ and corresponds to the situation of 0 member overlapping. $s_{1b}$ is the union of a subset of members of the maximal split, and violates the definition of maximal split.

If there are 2 members that overlaps with $s$, suppose they generate two different splitting vertex $v_a$ and $v_b$. Then $s_{ab}$ is a closed set containing $s_{xy}$ and corresponds to the situation of 0 member overlapping. $s_{ab}$ is the union of a subset of members of the maximal split, and violates the definition of maximal split.

If they generate the same splitting vertex $v_b$, from lemma 5, $v_b$ is also the endpoint vertex of at least 1 other member $s_{ab}$ which has to be inside $s$. Suppose the two overlapping members are $s_{cb}$ that contains $v_1$, and $s_{bd}$ that contains $v_2$. As the source vertex of $s$, $v_1$ has path to $v_b$ and $v_1$ has path to $v_a$, which implies $v_b$ has path to $v_a$. As the target vertex of $s$, $v_2$ has path from $v_b$ and $v_2$ has path from $v_a$, which implies $v_b$ has path from $v_a$. This conflicts with the fact that $s$ is acyclic. Therefore, this case is not possible.

Therefore, this lemma is proved.

$\square$

**Lemma 7.** *If non-branched $s_{ij}$ has at least 1 vertex but has 0 splitting vertex, then its maximal split has length $> 2$*

*Proof.* As $s_{ij}$ is not branched, the members of its maximal split cannot have the starting vertex as $v_i$ and the ending vertex as $v_j$ at the same time. If $s_{ij}$ has at least 1 vertex, and its maximal split has length 2, then its maximal split must be $\{[s_{ik}], [s_{kj}]\}$, and $v_k$ will be the splitting vertex of $s_{ij}$, which violates that $s_{ij}$ has no splitting vertex.

If $s_{ij}$ has at least 1 vertex without splitting vertex, it has at least 2 edges and cannot have a trivial length 1 maximal split. Therefore, its maximal split has length $> 2$ ☐

## A.2 Uniqueness of Division Tree

To prove this uniqueness, we simply discuss the division uniqueness of closed set type 1, 2 and 3.

### A.2.1 Uniqueness of Division of Closed Set Type 1

*Proof.* By the definition of this division and Lemma 4, the uniqueness of the division is equivalent to the uniqueness of the splitting vertex set of a closed set type 1. The splitting vertex set is obviously unique. ☐

### A.2.2 Uniqueness of Division of Closed Set Type 2

*Proof.* If there exists another division, there must be a branch member $s_{ij}^1$ in division 1 and a branch member $s_{ij}^2$ in division 2, where $s_{ij}^1 \cap s_{ij}^2 \neq \emptyset$ and $s_{ij}^1 \neq s_{ij}^2$.

Denote $s = s_{ij}^1 \cap s_{ij}^2$. By Lemma 1 and 2, $s = s_{ij}^3$ is also a closed set. As $s_{ij}^1$ and $s_{ij}^2$ cannot be divided into more branches, $s = s_{ij}^1 = s_{ij}^2$. Therefore, the division of closed set type 2 is unique. ☐

### A.2.3 Uniqueness of Division of Closed Set Type 3

*Proof.* As the closed set in the division tree has at least 1 vertex, with Lemma 7, we know that the division, i.e. maximal split of a closed set type 3 $s_{ij}$ within the division tree will have length $> 2$. Denote this maximal split as $\{[s_{pq}]\}$, we only need to prove this maximal split is unique.

Suppose there is a another different maximal split $\{[s'_{pq}]\}$, let us only check the difference between $\{[s_{pq}]\}$ and $\{[s'_{pq}]\}$. Denote $\{[s_{kt}]\}$ and $\{[s'_{kt}]\}$ with $\{[s_{pq}]\} - \{[s_{kt}]\} = \{[s'_{pq}]\} - \{[s'_{kt}]\}$ and $\nexists s \in \{[s_{kt}]\}, s' \in \{[s'_{kt}]\}, s = s'$. As $\{[s_{pq}]\} - \{[s_{kt}]\} = \{[s'_{pq}]\} - \{[s'_{kt}]\}$, we have $\cup\{[s_{kt}]\} = \cup\{[s'_{kt}]\}$

Obviously, $|\{[s_{kt}]\}| \geq 2$ and $|\{[s'_{kt}]\}| \geq 2$. Denote $\{v\}$ as all the endpoint vertices of $[s] \in \{[s_{kt}]\}$, and $\{v'\}$ for $\{[s'_{kt}]\}$. Obviously $\{v\} \neq \emptyset$ and $\{v'\} \neq \emptyset$. As $s_{ij}$ is non-branched, $\{v\} \cup \{v_i, v_j\} - \{v_i, v_j\} \neq \emptyset$ and $\{v'\} \cup \{v_i, v_j\} - \{v_i, v_j\} \neq \emptyset$.

Suppose $s_{ab}, s_{bc} \in \{[s_{kt}]\}$, according to Lemma 5, there's at least 1 other member that has $v_b$ as endpoint vertex. Suppose the other endpoint of this member is $v_d$. Let's discuss whether $v_b \in \{v'\}$ and whether $v_d \in \cup\{[s_{kt}]\}$.

If $v_d \notin \cup\{[s_{kt}]\}$, then $v_b$ must occur in $\{v'\}$. Otherwise, $v_b$ would be inside a closed set which would be violated by $v_d$. Given $v_b \in \{v'\}$, as $s_{ab} \notin \{[s'_{kt}]\}$, suppose $s_{eb} \in \{[s'_{kt}]\}$ and $s_{ab} \cap s_{eb} \neq \emptyset$. If $v_a \in s_{eb}$, from Lemma 1, $s_{eb}$ cannot be close. If $v_e \in s_{ab}$, from Lemma 5, $s_{ab}$ cannot be close. In this case, there cannot exist another different maximal split.

If $v_d \in \cup\{[s_{kt}]\}$, then $s_{bd} \in \{[s_{kt}]\}$. If $v_b \in \{v'\}$, we can use the same logic above to show this is impossible. Therefore, $v_b \notin \{v'\}$ and $s_{bd}$ is included by a closed set $s$. From Lemma 6, this is impossible. In this case, there cannot exist another different maximal split.

In all the cases, there cannot exist another different maximal split. Therefore, the maximal split is unique. ☐

## A.3 Completeness of Division Tree

Similar with the uniqueness, the completeness of division tree is equivalent to the completeness of the division of a closed set. To prove this completeness, we simply discuss the division completeness of closed set type 1, 2 and 3.

An equivalent statement of the division completeness is: there doesn't exist a closed set whose head is in one member of the division and whose tail is in another member of the division.

### A.3.1 COMPLETENESS OF DIVISION OF CLOSED SET TYPE 1

*Proof.* Suppose there exists a closed set $s$ whose head $v_p$ is in one member $s_1$ and whose tail $v_q$ is in another member $s_2$.

If $v_p$ is not an endpoint of $s_1$, then according to Lemma 3, $v_p$ is also a splitting vertex in $s_1$ and can break $s_1$ into smaller segments, which makes $v_p$ also the splitting vertex of the whole closed set. However, $v_p$ is not the splitting vertex of the whole closed set $s_{ij}$. This also applies to $v_q$. Therefore, the division of closed set type 1 is complete. □

### A.3.2 COMPLETENESS OF DIVISION OF CLOSED SET TYPE 2

*Proof.* Suppose there exists a closed set $s$ whose head $v_p$ is in one branch $s_{ij}^1$ and whose tail $v_q$ is in another branch $s_{ij}^2$. As $s$ crosses $s_{ij}^1$ and $s_{ij}^2$, there exists a boundary vertex $v$ in $s$, which belongs to $[s_{ij}^1]$ and has direct connection with a vertex outside $[s_{ij}^1]$. If $v$ is not $v_i$ or $v_j$, it will violate the independence of $s_{ij}$. If $v = v_i$, as $v_i$ is the head of both $s_{ij}^1$ and $s_{ij}^2$, it cannot be the boundary vertex, same when $v = v_j$. Therefore, there cannot exist such a closed set $s$. The division of closed set type 2 is complete. □

### A.3.3 COMPLETENESS OF DIVISION OF CLOSED SET TYPE 3

*Proof.* Suppose there exists a closed set $s$ whose head $v_p$ is in one member $s_1$ and whose tail $v_q$ is in another member $s_2$. Same with closed set type 2, the boundary vertex $v$ has to be the endpoint vertex of $s_1$ or the independence of $s_1$ will be violated. According to Lemma 5, $v$ is the endpoint vertex of at least 3 members, meaning that $v$ will at least have 1 connection with another closed set $s_3$. To maintain the independence of $s$, $s$ has to include $s_3$ as well. However, $s_3$ also has its endpoints. This will propagate until $s$ becomes the whole closed set. Therefore, there cannot exist such a closed set $s$. The division of closed set type 3 is complete. □

## B COMPLEXITY ANALYSIS

### B.1 ALGORITHM 1

Suppose there are $N$ vertices in the computation graph. There are $O(N^2)$ vertex pairs. For each vertex pair, the time cost is mainly on constructing accessibility graph and finding the shortest path. Denote the source vertex of the whole computation graph as $v_0$. To construct an accessibility graph, first we traverse the linear computation graph, record the accumulated sum $l(v_0, v_i)$ for each vertex $v_i$, and form a table of $l(v_i, v_j) = l(v_0, v_j) - l(v_0, v_i) - l(v_i)$. These steps will cost $O(N^2)$. Then we traverse each $(v_i, v_j)$ pair to form the edges of the accessibility graph, which also cost $O(N^2)$. Solving the shortest path problem in accessibility graph will also cost $O(N^2)$ as the accessibility graph has $N$ vertices. Therefore, the overall time complexity of Algorithm 1 would be $O(N^4)$.

The space complexity would be $O(N^2)$ for the table of $l(v_i, v_j)$ and the accessibility graph itself.

### B.2 ALGORITHM 2

Suppose there are $N$ vertices in the closed set $s_{ij}$. In step 1, getting $\{v_{in}\}$ and $\{v_{out}\}$ will cost $O(N)$ time for traversing the ancestors and descendents of $v_t$. In our implementation, an array $a$ of length $N$ is used to represent $\{v_{in}\}$ and $\{v_{out}\}$: $a_i = 1$ indicates $v_i \in \{v_{in}\}$, $a_i = 2$ indicates $v_i \in \{v_{out}\}$ and $a_i = 0$ indicates $v_i \notin \{v_{in}\} \cup \{v_{out}\}$. Then the union check and intersection check in step 2 can be done in $O(N)$. The connection check in step 2 traverses the edges and costs $O(N^2)$. Other steps are $O(1)$. Therefore, the overall time complexity of Algorithm 2 would be $O(N^2)$.

The space complexity would be $O(N)$ for the array to represent $\{v_{in}\}$ and $\{v_{out}\}$.

### B.3 ALGORITHM 3

Suppose there are $N$ vertices in the closed set $s_{ij}$. The most time consuming part will be from step 5 to step 13. Other steps are $O(1)$. In step 5 to step 13, every edge between two vertices in $s_{ij}$ is at most visited once and there are $O(N^2)$ edges. Therefore, the overall time complexity of Algorithm 3 would be $O(N^2)$.

In our implementation, an array of length $N$ is used to represent the vertex set $s = \{v_k\}$. Therefore, the space complexity would be $O(N)$.

### B.4 ALGORITHM 4

Suppose there are $N$ vertices in the closed set $s_{ij}$ and there are $O(N^2)$ vertex pairs. For each vertex pair, the connection check in step 2-4 will cost $O(N^2)$, similar to the connection check in Algorithm 2. Thus step 1-4 will cost $O(N^4)$. In our implementation, for each vertex in the closed set $s_{ij}$, we select the largest formed closed set $s_{kt}$ that contains this vertex. The closed set number is then reduced to $O(N)$ and step 5-6 can be done in $O(N^3)$. Therefore, the overall time complexity of Algorithm 4 would be $O(N^4)$

As $O(N^2)$ closed sets can be formed in step 1-4 and each closed set is a smaller DAG with $O(N)$ vertices and cost $O(N^2)$ space, the space complexity would be $O(N^4)$ for all these closed sets.

### B.5 ALGORITHM 5

Step 1 is similar to step 1-4 in Algorithm 4 with $s_{ij}$ being the whole computation graph. Therefore, the overall time complexity for step 1 is $O(N^4)$.

In step 2, the complexity of building division tree is related to the complexity of getting the division of a closed set. For closed set type 1, Algorithm 2 is called for each vertex to get all splitting vertices. Thus getting the division of closed set type 1 cost $O(N^3)$ time. For closed set type 2, Algorithm 3 is used to solve for its division and costs $O(N^2)$ time. For type 3, Algorithm 4 is called to solve for its division. Notice that we have already stored all possible closed sets in step 1, step 1-4 in Algorithm 4 can be skipped and thus the time complexity of getting the division of closed set type 3 is reduced to $O(N^3)$. Therefore, getting the division of an arbitrary closed set costs $O(N^3)$ time. In depth $i$ of the division tree, suppose there are $k$ closed sets, and the number of vertices of $j$th closed sets is $a_j$. To build depth $i + 1$ of the division tree, we need to get the division of all these closed sets, which will cost $\sum_j O(a_j^3)$. As $\sum_j a_j \leq N$, we have $\sum_j O(a_j^3) \leq O(N^3)$. As the depth of division tree is at most $N$, the overall time complexity of step 2 would be $O(N^4)$.

For step 3-10, if the computation graph is linear, the ACG solver will reduce to LCG solver and has complexity $O(N^4)$. If the computation graph is non-linear, the length of $\{m\}$ would be $O(N^2)$ for there are $O(N^2)$ vertex pair. For a max term $m$, from step 4-10, the actual time costing part will be step 6 which calls the LCG solver, and other steps would be $O(1)$. Suppose the LCG solver is called $k$ times, solving problems of $a_1, a_2, ..., a_k$ vertices. The total complexity of this would be $O(a_1^4) + O(a_2^4) + ... + O(a_k^4)$. Notice that $a_1 + a_2 + ... + a_k \leq N$ for the fact that any vertex in the computation graph would not be solved twice by LCG solver, we have $a_1^4 + a_2^4 + ... + a_k^4 \leq N^4$. Therefore the time complexity of step 3-10 is $O(N^4)$.

Step 1 would cost $O(N^4)$ space to store all the possible closed sets. Step 2 would cost $O(N^2)$ space for the division tree. Step 3-10 would cost $O(N^2)$ space for calling LCG solver.

In conclusion, the overall time complexity of Algorithm 5 is $O(N^4)$ and the overall space complexity of Algorithm 5 is $O(N^4)$.

Table 2: Actual Runtime of ACG Solver and Theoretical Analysis

| Linear network | Number of vertices | Runtime (s) | Measured Memory Cut off | Theoretical Memory Cut off |
|---|---|---|---|---|
| Alexnet | 12 | 0.03 | 26% | 42% |
| Vgg11 | 17 | 0.09 | 39% | 50% |
| Vgg13 | 19 | 0.15 | 38% | 47% |
| Vgg16 | 22 | 0.26 | 42% | 51% |
| Vgg19 | 25 | 0.44 | 48% | 53% |
| Non-linear network | Number of vertices | Runtime (s) | Measured Memory Cut off | Theoretical Memory Cut off |
| Resnet18 | 51 | 0.09 | 46% | 63% |
| Resnet34 | 91 | 0.53 | 60% | 73% |
| Resnet50 | 125 | 1.27 | 65% | 75% |
| Resnet101 | 244 | 12.40 | 75% | 81% |
| Resnet152 | 363 | 59.34 | 80% | 84% |
| Densenet121 | 306 | 293.75 | 81% | 81% |
| Densenet161 | 406 | 1537.82 | 83% | 84% |
| Densenet169 | 426 | 2356.47 | 82% | 84% |
| Densenet201 | 506 | 7335.35 | 82% | 86% |
| Inceptionv3 | 219 | 7.68 | 69% | 76% |
| Custom | 35 | 0.05 | 58% | 68% |

## C  RUNTIME AND THEORETICAL ANALYSIS

The number of vertices in the computation graph and the runtime of ACG Solver (Algorithm 5) for each network are listed in Table 2. All the runtimes were measured on a single core of CPU i7-8700.

Notice that the runtime is measured on only 1 cpu core, it can be massively reduced by parallelization on multiple cpu cores. The runtime can also be further reduced through a better implementation as our implementation is a prototype.

Although it might be concerning that the runtime is too much for some deep networks, it is still relatively small compared to training processes which might cost days or even weeks. More importantly, solving the optimal solution for a network is an one-time effort. The optimal solutions for all popular networks will be released online for people to use without taking the time to run ACG solver.

To see how well the reality matches with the theory, we also compare the measured memory cut off and theoretical memory cut off (given by Algorithm 5) in Table 2. Observe that all the measured memory cut off are slightly lower than theoretical memory cut off. This is because, in implementation, we assume that the whole input tensors of each operation are always stored for backward. In reality, some operations only need to store small tensors for backward. For example, batch-normalization only needs a few statistics for backward and doesn't need the whole input tensor.

## D  VISUALIZATION

We visualize the computation graph of Alexnet, vgg11, vgg13, vgg16 ,vgg19 and CustomNet and the solution of our approach (in green) and the solution of Chen's approach (in red). In the computation graphs, the cost of each vertex and the actual operation of each edge are also marked. The cost of each vertex is the size of this tensor during forward given the input as $[1, 3, 224, 224]$ ($[1, 3, 300, 300]$ for inception v3). For example, in Alexnet, the input is $[1, 3, 224, 224]$ and thus the source vertex has the cost $150528 = 1 \times 3 \times 224 \times 224$. After 2D convolution and relu, the tensor becomes $[1, 64, 55, 55]$ and thus the second vertex has the cost $193600 = 1 \times 64 \times 55 \times 55$.

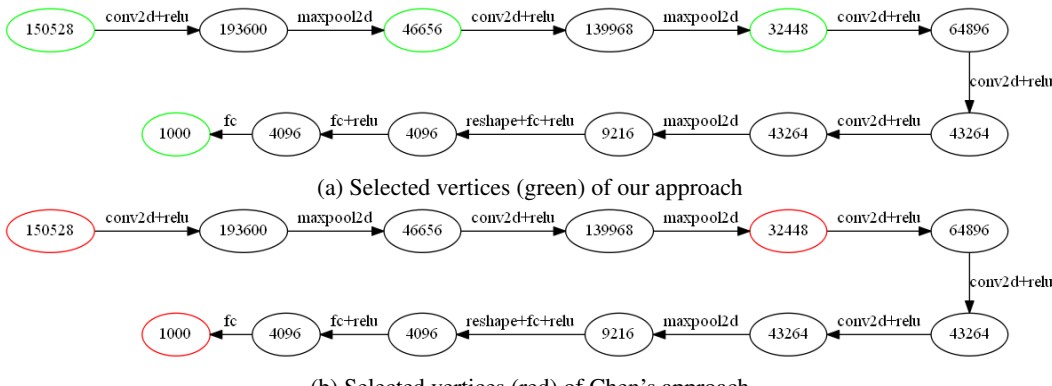

(a) Selected vertices (green) of our approach

(b) Selected vertices (red) of Chen's approach

Figure 6: Endpoint vertices found on Alexnet

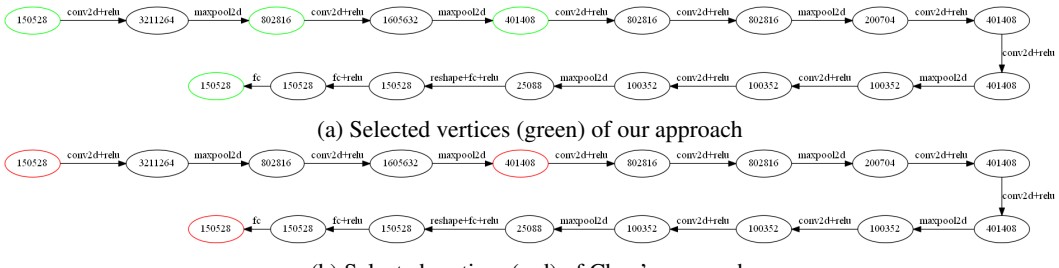

(a) Selected vertices (green) of our approach

(b) Selected vertices (red) of Chen's approach

Figure 7: Endpoint vertices found on vgg11

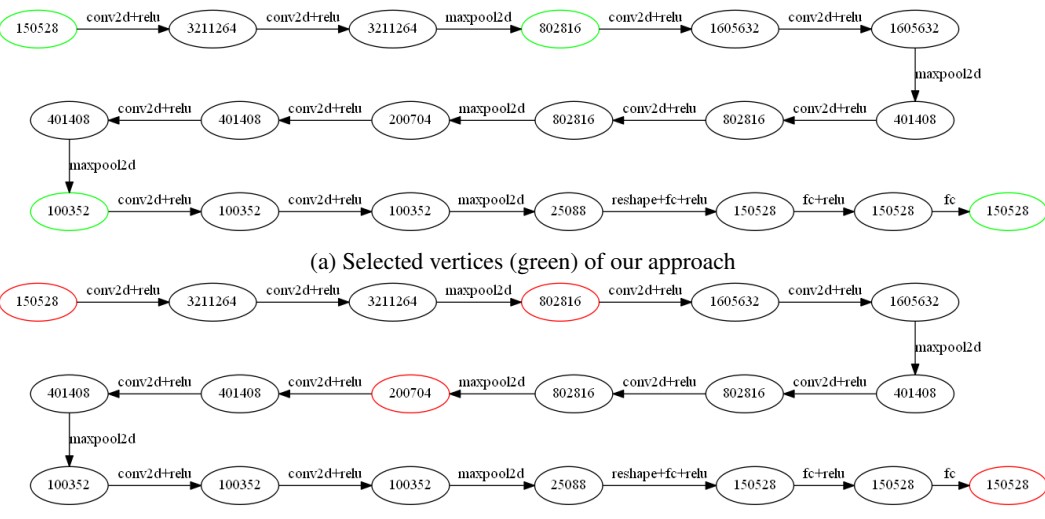

(a) Selected vertices (green) of our approach

(b) Selected vertices (red) of Chen's approach

Figure 8: Endpoint vertices found on vgg13

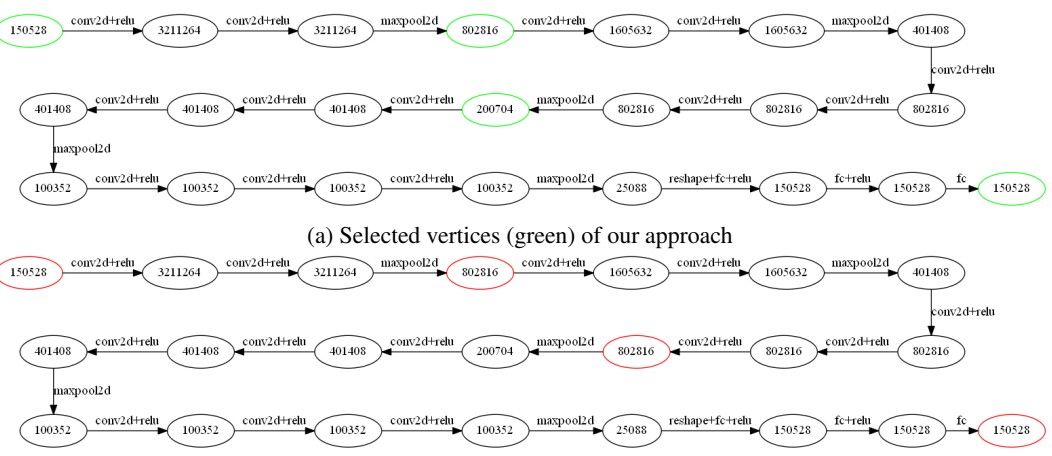

(a) Selected vertices (green) of our approach

(b) Selected vertices (red) of Chen's approach

Figure 9: Endpoint vertices found on vgg16

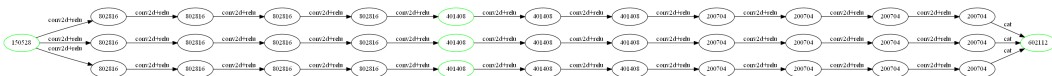

Figure 10: Endpoint vertices found on CustomNet, Selected vertices(green) of our approach, Chen's approach not applicable

