# OpenReview forum: "Cutting Down Training Memory by Re-fowarding"
_ICLR.cc/2019/Conference_

### Official Review · AnonReviewer3 · 2018-11-01
**Nice and simple**

**Rating:** 6
**Confidence:** 3

**Review:**

Summary: The paper suggests a method to reducing the space consumption of training neural nets, in exchange for additional training time. The method stores in memory only a subset of the intermediate tensors computed in the forward step, and then in the backward step it re-computes the missing tensors as they are needed by interpolating forward (again) between the stored tensors. The paper also gives a combinatorial algorithm for choosing which tensors to store on a given computation DAG annotated with vertex costs.

Evaluation: I generally like the paper. The proposed method is simple and straightforward, and seems to lead to a noticeable improvement in space usage during training.
The part related to decomposing a DAG into "close sets" looks like it might overlap with existing literature in graph theory; I don't have concrete references but the authors may want to check this. The ACG solver algorithm looks somewhat wasteful in terms of the degree of the polynomial running time, but since actual computation graphs are tiny in computational terms, I guess this is not really an issue.
One point not addressed in the experiments is what is the overhead incurred in training time by the two space-efficient methods over usual training. I suppose one expects the training time be less than twice longer, since the re-forwards amount to one additional complete forward step per forward-backward pair.
Another thing that would be interesting to see is the actual stored vertices (V_R) that were chosen empirically for the rows in table 1 (or at least some rows). Since the computational graphs of the tested networks are small, and some are well-known, I imagine it should doable (though this is merely a suggestion).

Conclusion: The paper is simple and looks reasonable, with no major issues that I could detect.

---

> ### Author Response · Authors · 2018-11-26
> **Response to Reviewer 3**
>
> Question: “The part related to decomposing a DAG into "closed sets" looks like it might overlap with existing literature in graph theory”
>
> Answer:  To the best of our knowledge, there is no similar definition of closed set in existing literature.
>
> Question: “The ACG solver algorithm looks somewhat wasteful in terms of the degree of the polynomial running time, but since actual computation graphs are tiny in computational terms, I guess this is not really an issue.”
>
> Answer: You are absolutely right. For instance for big DNNs such as Resnet101, there are only 244 vertices. We have added a new section B in the appendix for detailed complexity analysis. The overall complexity of ACG solver is O(N^4) with N being the number of vertex in the computation graph. We have also added a new section C in the appendix to show vertex number and the actual runtime of the ACG solver.
>
> Question: “One point not addressed in the experiments is what is the overhead incurred in training time by the two space-efficient methods over usual training.”
>
> Answer: The time overheads of the two space-efficient methods are the same. Both methods cost one additional complete forward per forward-backward integration. We have measured the training time for one iteration, and added three columns: the regular training time, space efficient training time and relative overhead to Table 1. We have also changed the column of Measured Memory Ratio to Memory Cut off to allow a straightforward comparison between memory cut off and time overhead.
>
> Question: “Another thing that would be interesting to see is the actual stored vertices (V_R) that were chosen empirically for the rows in table 1 (or at least some rows)”
>
> Answer: We have added a new section D in the appendix to show some stored vertices of our approach and Chen’s approach.

---

### Official Review · AnonReviewer1 · 2018-11-04
**Manuscript is rough -- suggestion is to reject.**

**Rating:** 4
**Confidence:** 2

**Review:**

In Cutting Down Training Memory by Re-forwarding the authors present a method for structuring the computation graph of a DNN to save training time.  The authors given experiments that show that the method can save up to 80% training memory.

The idea paper is nice however, this draft needs more writing work to bring to a conference standard in my opinion.

My objection to the writing begins with the Definition and example of a “Close set” which I details below.

I also have further suggestions:


Questions & doubts

Definition 1 : Close set

* A set of vertices and edges that start from v_i and that end at  v_j.

1. Vertices don’t start or end at vertices so I am already confused
2. No notation has been introduced yet for close set so I am not sure what i and j refer to.  Also I suspect you want “or” and not “and” otherwise there could only be one possible “edge” (i,j)

* sij = {v,e} is called a close set such that ∀v1 ∈ sij, v1 has no edge to any v2 \not\in sij ∪ {vi , vj }

1. Do you mean this is a set with two elements v and e?  Probably not?
2. Is v1 and v2 meant to be generic vertices?  If so it quite unnatural to also use v_i and v_j.  I.e., probably the name of two specific vertices in your graph is “v1” and “v2”
3. Another question is the close set s_{ij} unique or are their potentially multiple s_{ij}?

* The edges of sij are all the edges between v1 , v2 ∈ sij , all the edges between vi and v ∈ sij

1. Not getting what’s going with v1 and v2 versus vi and vj.

Examples of close set
a Probably you should phrase as there can exist a close s_{2,4} since …

After reading this I got general idea that close set correspond to connected component with no connections except to oldest ancestor and youngest child.  But that is a guess — the notation and precision in the definition as well as the examples led me to have too many doubts.

Also

P3 Case(1) is n and N the same?

With respect to algorithm 5 Can you discuss its computation time?

In summary this is potentially interesting work but the writing should be sharper, their should be less ambiguity of interpretation of close set.

I should note — this reviewer lacks confidence in his review in so far as they have next to zero experience with DNNs.  So if given the problems in the manuscript the contribution paper would be for example a key result for DNN training this reviewer would not be able to recognize it as such.

---

> ### Author Response · Authors · 2018-11-26
> **Response to Reviewer 1 Part 1**
>
> Question:  “The definition of close set.”
>
> Answer: Admittedly, the original definition of “close set” may cause confusing. We have rewritten the definition as “closed set” and explained below.
>
> As described in section 4.1, we divide an Arbitrary Computation Graph (ACG) into segments at a subset of vertices. We call these vertices as the endpoint vertices of the segments. We only deal with the segment with two endpoint vertices, namely one source vertex v_i and one target vertex v_j. A closed set is defined on the set of vertices and edges belonging to a segment. For example, in figure 3(a), a set of  vertices and edges between source vertex v1 and target vertex v3 is denoted as s={v2, e_{12}, e_{23}, e_{13}}.
>
> Closed Set Definition: A set of vertices and edges, s,  is a closed set if and only if it satisfies the following three properties: 1. All the vertices of s have a common ancestor v_i and a common descendent v_j.  2 Denote the vertex subset of s as V, the edge subset as E, the set of edges between two arbitrary vertices of V union {v_i, v_j} as E^\prime, and the edge from v_i to v_j (if exists) as e_{ij}. E must either be E^\prime or E^\prime - {e_{ij}};. (3) An arbitrary vertex v1\in V doesn't have edge with another arbitrary v2 outside of V union {v_i, v_j}. For multiple valid closed sets between v_i and v_j, we denote the largest one as s_{ij}.
>
> Property 1 confines a closed set on segments with single source vertex v_i and single target vertex v_j. Property 2 confines the edge subsets of s to be one of two cases: E^\prime or E^\prime-\{e_{ij}\}. Property 3 guarantees that the vertices within s have no connections with other vertices outside s union {v_i, v_j}. As there might be multiple valid closed sets between v_i and v_j, which corresponds to the Branched Closed Set in Definition 5, we denote the largest closed set between v_i and v_j as s_{ij} and denote smaller closed set with an extra superscript, such as s_{ij}^1.
>
> The remaining of your questions are answered as follows
>
> Question: “A set of vertices and edges that start from v_i and that end at v_j. 1. Vertices don’t start or end at vertices so I am already confused 2. No notation has been introduced yet for closed set so I am not sure what i and j refer to. Also I suspect you want “or” and not “and” otherwise there could only be one possible “edge” (i,j)”
>
> Answer: 1. By saying vertexes start from v_i and end at v_j, we mean vertices which have v_i as a common ancestor and v_j as a common descendant. These vertices have paths from v_i and have paths to v_j).
> 2. Given a set s of vertices and edges, v_i and v_j are two certain vertices of this set (namely the two endpoints) that satisfies property 1, 2 and 3 in the new definition.
>
>
> Question: “sij = {v,e} is called a closed set such that ∀v1 ∈ sij, v1 has no edge to any v2 \not\in sij ∪ {vi , vj } 1. Do you mean this is a set with two elements v and e? Probably not? 2. Is v1 and v2 meant to be generic vertices? If so it quite unnatural to also use v_i and v_j. I.e., probably the name of two specific vertices in your graph is “v1” and “v2” 3. Another question is the closed set s_{ij} unique or are their potentially multiple s_{ij}?”
>
> Answer: (1) The set s_{ij} is a union of vertices and edges, meaning that the element of this set can be either vertex or edge.  A subset of s_{ij} can be a vertex set (if no edge in it), an edge set (if no vertex in it) or still a union of vertices and edges.
> (2) v_1 and v_2 are two arbitrary vertices in the closed set s_{ij}.
> (3) There can be multiple closed sets between a source vertex vi and a target vertex v_j. For example in Figure 4(b), there are 4 valid closed sets between vertex 1 and vertex 2: the closed set of all the vertex and edges; the closed set of the upper branch; the closed set of the middle branch; the closed set of the lower branch. In this paper, s_{ij} always denotes the largest closed set between v_i and v_j. When needed, we use an extra superscript to describe the subset of s_{ij}, such as s_{ij}^1, s_{ij}^2.

---

> > ### Author Response · Authors · 2018-11-26
> > **Response to Reviewer 1 Part 2**
> >
> > Question: “The edges of sij are all the edges between v1 , v2 ∈ sij , all the edges between vi and v ∈ sij 1. Not getting what’s going with v1 and v2 versus vi and vj.”
> >
> > Answer:  For the  closed set s_{ij}, vi and vj are the endpoint vertices. v1 and v2 are two arbitrary vertices within the  closed set s_{ij}.
> >
> > Question: “Examples of closed set a Probably you should phrase as there can exist a close s_{2,4} since...”
> >
> > Answer:  Thank you for pointing out this issue. We have fixed it in the revised paper.
> >
> > Question: “P3 Case(1) is n and N the same?”
> >
> > Answer:  n and N is the same. Sorry for the typo. We have replaced all the “n”s with “N”s in the revised paper. Thank you for pointing it out!
> >
> > Question: “With respect to algorithm 5, Can you discuss its computation time?”
> >
> > Answer: The complexity of Algorithm 5 is O(N^4) with N being the number of vertices in the computation graph. For instance for Resnet101, N = 244. We have added section B in appendix for detailed complexity analysis. We have also included a table of vertex number and actual runtime of Algorithm 5 in section C in appendix.

---

### Official Review · AnonReviewer5 · 2018-11-12
**Incremental improvements. Latency overhead is not reported.**

**Rating:** 4
**Confidence:** 3

**Review:**


This paper introduces a re-forwarding method to cut the memory footprint for training neural networks. It improved on top of previous work that did re-forwarding for linear computation graph, and this work also supports non-linear computation graph.

Pros:
(1) This paper is solving an important problem saving the training memory of deep neural networks. GPU price and GPU memory increase non-proportionally, and many machine learning tasks require large GPU memory to train.  Though, there's rich literature in the area of saving the GPU memory for training.

(2) This paper evaluated a large number of neural networks (Alexnet, VGG, ResNet, DenseNet). The evaluation is extensive.

Cons:
(1) this paper has incremental contribution compared with Chen et al. (2016). Chen et al worked on linear computation graph. Extending the algorithm from linear graph to non-linear graph is important but the technical contribution is incremental and thin.

(2) the improvement over previous work is quite marginal. For example, the memory saving went from 2894MB to 2586MB for VGG16, from 2332Mb to 1798MB for ResNet-50, just to pick a few widely used architectures.

(3) The paper only reported the memory saving, but didn't report the latency overhead. It's important to know how much latency overhead is brought by this algorithm. If we save 2x memory but the training time also get much longer, users will simply reduce the virtual batch size by half and do forward the backward twice to simulate the same physical batch size. So, the authors should be clear how much training time is brought by such algorithm and report the *end-to-end training time* in Table 1, with and without re-forwarding.

---

> ### Author Response · Authors · 2018-11-26
> **Response to Reviewer 5**
>
> Question: “This paper has incremental contribution compared with Chen et al. (2016). Chen et al worked on linear computation graph. Extending the algorithm from linear graph to non-linear graph is important but the technical contribution is incremental and thin.”
>
> Answer: We cannot agree to this assertive remarks. Comparing to Chen et al. (2016)’s heuristic solution for only linear computation graphs, our work achieved optimal solutions for both linear non-linear computation graphs. This is a significant contribution to modern deep neural networks.
>
> Otherwise, to save memory on nonlinear networks, one may need to follow our synthetic steps in experiments: Use our Algorithm 2 to find splitting vertices, MANUALLY re-organized all the non-linear computation graphs into linear computation graphs, and fed them to Chen et al. (2016). Moreover, for nonlinear networks such as the “CutomNet” in Table 1, even with above manual operations, Chen et al. (2016) does not apply.
>
>
> Question: “The improvement over previous work is quite marginal. For example, the memory saving went from 2894MB to 2586MB for VGG16, from 2332Mb to 1798MB for ResNet-50, just to pick a few widely used architectures.”
>
> Answer: Please note that (1) Chen’s approach is not optimal and performance can vary a lot on different networks. Our approach is optimal and hence more reliable. There are many cases where gpu memory has become very intensive, such as high resolution image segmentation and video object tracking. For example, in semantic segmentation, the image of Cityscapes dataset is of 1024*2048, a single image would consume 6932 MB on Resnet 50 and more than 10000 MB on Resnet 101 in regular training approach. In the task of video object tracking, given the input as a video sequence of 50*3*300*300, resnet50 consumes 7642 MB and Resnet101 consumes more than 12000 MB. To cut down memory in these cases, surely an optimal solution is a lot more favorable than a non-optimal solution.
>     (2) Chen et al. (2016) only worked on linear computation graphs and DOES NOT work on popular nonlinear networks, such as Resnet.  To compare on non-linear networks, we used our Algorithm 2 to find splitting vertices, MANUALLY re-organized all the non-linear computation graphs into linear computation graphs, and fed them to Chen et al. (2016). This is an adaptation we made for Chen’s approach to allow comparison on non-linear networks. In other words, if a computation graph doesn’t have splitting vertexes, adapted Chen’s approach can’t work, while our Re-forwarding approach directly works on nonlinear networks and solves for optimal solution. To show this, we have added a customized network "CustomNet" to Table 1 and visualized its computation graph in section D. On this customized network, Chen’s approach is not applicable even after our adaptation, and our approach can still cut down significant amount of memory.
>
>
> Question: The paper only reported the memory saving, but didn't report the latency overhead.
>
> Answer: Thanks for pointing this out. The time overheads of both two Chen and our method involve one additional complete forward per forward-backward iteration.  We have added the regular training time (time for 1 iteration), space efficient training time and relative overhead to the result table in the experiment section. We have also changed the column of “Measured Memory Ratio” to “Memory Cut off” to allow a straightforward comparison between memory cut off and time overhead.

---

### Official Review · AnonReviewer4 · 2018-11-12
**An intuitive idea but execution can be more convincing**

**Rating:** 6
**Confidence:** 3

**Review:**

I took a look at the revision.  I am glad to see that the authors clarified the meaning of "optimality" and added time complexity for each algorithm. The complexities of the algorithms do not seem great (3rd or 4th order polynomial of N) as they appear to be checking things exhaustively, but perhaps they are not a big issue since the decomposition algorithm is run only once and usually N is not huge. I wonder if more efficient algorithms exist.

It is also nice to see that the time overhead for one training step is not huge (last column of Table 1). While I still think it is better to see more complete training curves, the provided time overhead is a proxy.

I hope the authors can further improve the algorithm description, for example, the pseudo-code of Algorithm 1 is very verbal/ambiguous, and it would be better to have more implementation-friendly pseudo-code.

Despite the above-mentioned flaws, I think this work is still valuable in handling the memory consumption of arbitrary computation graph in a principled manner.

=============================================================================
This paper presents a method for reducing the memory cost of training DNNs. The main idea is to divide the computational graph into smaller components (close sets), such that the dependency between components is low, and so one can store only tensors at key vertices in each component during the forward pass. In the backward pass, one needs to re-forward the data within each close set for computing the gradient.

The idea is quite intuitive and the example in linear computational graph in Section 3 clearly demonstrates the basic idea. The main technical development is in the case of arbitrary computation graph (Section 4), where the authors explain how to divide the computational graph into different types of close sets. I have not read the proofs in appendix, but it is clear that the technical development is mainly in discrete math rather than machine learning. For this part, my suggestions are:
1. give better algorithm description: what are the inputs and outputs of each algorithm, for the examples in Figure 2-4 (and perhaps with costs for each vertex), which vertices do the final algorithm decide to store
2. analyze the complexity of each algorithm
3. provide clear definition of "optimality" and its proof: the authors mentioned in a few places about the method being "optimal", but the paper needs to be crystal clear about "for what problem is the method optimal", "optimal in what sense (the objective)".

The more concerning part is the experimental evaluation. While I believe that the proposed method can save memory cost, there is no result on the additional time cost for re-forwarding. Therefore, it is not clear if it is worth the extra complication of using re-forwarding in an end-to-end sense: the authors motivated in Section 1 that saving memory cost allows one to use larger mini-batches, and an important benefit of large mini-batch is to accelerate training, see, e.g.,
Goyal et al. Accurate, large minibatch SGD: Training imagenet in 1 hour.
Hoffer et al. Train longer, generalize better: Closing the generalization gap in large batch training of neural networks.
It is less desirable to use re-forwarding it if it causes significant latency in the backward pass and training. An good demonstration of the memory vs. time tradeoff would be the training curves of loss vs. running time.

The other detail I would appreciate is how the authors implemented the algorithm, and whether the proposed method is likely to be deployed on popular ML frameworks, which would improve the significance of the work.

---

> ### Author Response · Authors · 2018-11-26
> **Response to Reviewer 4**
>
> Question: “Give better algorithm description: what are the inputs and outputs of each algorithm, for the examples in Figure 2-4 (and perhaps with costs for each vertex), which vertices do the final algorithm decide to store.”
>
> Answer: The inputs of Algorithm 1 and 5 are the whole computation graph described by its vertices and edges. The outputs are the endpoint vertices that achieve the optimal memory saving. Specifically, the inputs of Algorithm 2 are a closed set s_{ij} and a vertex v_t and the output is a boolean value indicating whether v_t is a splitting vertex of s_{ij}. The input of Algorithm 3 is a closed set s_{ij} and the output is a boolean value indicating whether s_{ij} is branched. The input of Algorithm 4 is a type 3 closed set s_{ij} and the output is its division.
>    We have added a new section D in the appendix to show some stored vertices of our approach and Chen’s approach. In section D, we also explained how the cost of each vertex is determined.
>
> Question: “Analyze the complexity of each algorithm”
>
> Answer: We have added complexity of each algorithm in the revised paper. Detailed complexity were added to Appendix section B and actual runtime of ACG Solver (Algorithm 5) for each network were added in section C. The time complexities of Algorithm 1 and 5 are both O(N^4), with N being the number of vertices in the computation graph. The time complexity of Algorithm 2 is O(N^2), of Algorithm 3 is O(N^2) and of Algorithm 4 is O(N^4) with N being the number of vertices in the closed set s_{ij}.
>
> Question: “Provide clear definition of "optimality" and its proof: the authors mentioned in a few places about the method being "optimal", but the paper needs to be crystal clear about "for what problem is the method optimal", "optimal in what sense (the objective)"”
>
> Answer: The problem to be optimized is: Finding the best decomposition of the computation graph into multiple segments, such that storing the endpoint vertices of these segments has the lowest memory cost (Equation (1) in section 3). Note that this decomposition is only dependent on the network structure and is independent from the forward and backward operations.
>    The optimality of our method is achieved under two assumptions (explained in section 4.1): (1) the produced segments only has two endpoints (namely one source vertex and one target vertex). (2) the operation with multiple inputs is able to pass the gradient of output to its inputs without relying on the concurrent value of inputs.
>
> Question: “While I believe that the proposed method can save memory cost, there is no result on the additional time cost for re-forwarding”
>
> Answer: The time overheads of the two space efficient methods are the same, namely one additional complete forward per forward-backward pair. We have added the regular training time (time for 1 iteration), space efficient training time and relative overhead to the result table in the experiment section. We have also changed the column of “Measured Memory Ratio” to “Memory Cut off” to allow a straightforward comparison between memory cut off and time overhead.
>
> Question: “The other detail I would appreciate is how the authors implemented the algorithm, and whether the proposed method is likely to be deployed on popular ML frameworks”
>
> Answer:  For the experiment, we implemented the algorithm in Pytorch. We developed a wrapper class for pytorch models and implemented the algorithms as class functions. We will first release the Pytorch version of our algorithms and several pre-computed solutions for popular networks. In the future, we will also deploy this approach on other popular ML frameworks, such as tensorflow, Keras and Caffe.

---

### Author Response · Authors · 2018-11-26
**Summary of Paper Changes**

We thank all reviewers for their insightful comments and the efforts to help us improve the quality of our paper. Below are the changes we made to our paper.

1. In the revised paper, the abstract is slightly modified.

2. In the revised paper, the “vertexes” is updated as “vertices”.

3. In the revised paper, the “close set” is updated and clarified as “closed set”.

4. In the revised paper, time complexity is added to the explanation of each algorithm.

5. In section 3 Case (1),“n” is changed to “N”

6. In section 4.2, the description of figure 3 is slightly modified.

7. In section 4.2, Definition 1 and its explanation are rewritten.

8. In section 5, three columns are added to Table 1: regular training time (time for 1 iteration), space efficient training time and relative overhead. The column of “Measured Memory Ratio” is changed to “Memory Cut off” for a straightforward comparison between memory saving and time overhead. The column of theoretical memory usage ratio is moved to appendix. We added a new row “CustomNet” as an example where Chen’s method does not apply even with its manual implementation. The content of section 5 is also modified accordingly.

9. In the Appendix, section B is added for detailed complexity analysis.

10. In the Appendix, section C is added to show actual runtime of ACG solver and theoretical analysis.

11. In the Appendix, section D is added for visualization of some computation graphs.

---

### Meta-Review · Area_Chair1 · 2018-12-17
**borderline**

**Confidence:** 3
**Recommendation:** Reject

**Metareview:**

This paper proposed a method to reduce the memory of training neural nets, in exchange for additional training time. The paper is simple and looks reasonable. It's a natural followup with previous work.  The improvement over previous work is not significant, with some overhead incurred in training time. This is a borderline paper but given the <30% acceptance rate, I need to downgrade the paper to reject.